# Causes of Admission, Mortality and Pathological Findings in European Hedgehogs: Reports from Two University Centers in Italy and Switzerland

**DOI:** 10.3390/ani14131852

**Published:** 2024-06-22

**Authors:** Ilaria Prandi, Eva Dervas, Elena Colombino, Giuseppe Bonaffini, Stefania Zanet, Riccardo Orusa, Serena Robetto, Massimo Vacchetta, Mitzy Mauthe von Degerfeld, Giuseppe Quaranta, Udo Hetzel, Maria Teresa Capucchio

**Affiliations:** 1Centro Animali Non Convenzionali, Department of Veterinary Sciences, University of Turin, 10095 Grugliasco, Italy; giuseppe.bonaffini@unito.it (G.B.); mitzy.mauthe@unito.it (M.M.v.D.); giuseppe.quaranta@unito.it (G.Q.); mariateresa.capucchio@unito.it (M.T.C.); 2Department of Veterinary Sciences, University of Turin, 10095 Grugliasco, Italy; elena.colombino@edu.unito.it (E.C.); stefania.zanet@unito.it (S.Z.); 3Institute of Veterinary Pathology, Vetsuisse Faculty, University of Zurich, 8057 Zurich, Switzerland; eva.dervas@uzh.ch (E.D.); udo.hetzel@uzh.ch (U.H.); 4Patología y Sanidad Animal, Departamento Producción Sanidad Animal, Salud Pública Veterinaria y Ciencia y Tecnología de los Alimentos, Facultad de Veterinaria, Universidad CEU Cardenal Herrera, CEU Universities, 46115 Alfara del Patriarca, Spain; 5National Reference Centre for Wild Animals Diseases (CeRMAS), S.C. Valle d’Aosta, Istituto Zooprofilattico Sperimentale del Piemonte, Liguria e Valle d’Aosta (IZSPLV), 11020 Quart, Italy; riccardo.orusa@izsto.it (R.O.); serena.robetto@izsto.it (S.R.); 6Centro Recupero Ricci “La Ninna”, 12060 Novello, Italy; ninnaeisuoiamici@libero.it; 7National Research Council of Italy-Institute of Sciences of Food Production (CNR-ISPA), 10095 Grugliasco, Italy

**Keywords:** European hedgehogs, *Erinaceus europaeus*, causes of mortality, pathological findings, verminous pneumonia, wildlife rescue centers

## Abstract

**Simple Summary:**

A decline in European hedgehog populations has been reported in several European countries over the past years. Human activities, along with global warming and infectious diseases, appear to be the main drivers of this phenomenon. The present study aimed to determine the major causes of mortality and the main pathological findings in European hedgehogs (*Erinaceus europaeus*) submitted to postmortem examination at two institutes of veterinary pathology, one located in Italy (Piedmont region) and one in Switzerland (Canton of Zurich). Hedgehogs were found in affected health status in the wild, mostly due to traumatic injury, poor overall condition and respiratory, gastrointestinal or neurological signs. The primary causes of death identified were infectious diseases and traumatic insults. The lungs were the most commonly affected organ, mainly displaying pneumonia, which was significantly associated with the presence of nematodes. These findings should be held in mind when treating the species presented to wildlife rescue centers and highlight the importance of proper education of the public on how to coexist and interact with hedgehog populations inhabiting urban areas.

**Abstract:**

European hedgehogs (*Erinaceus europaeus*) are nocturnal insectivores frequently found in urban areas. In the last decades, their population has declined in various European countries and human activities have emerged as significant contributors to this trend. While the literature has mainly focused on trauma as the major cause of mortality, few authors have considered pathological findings. The present study is based on the results of full post-mortem examinations performed on 162 European hedgehogs in Italy and 109 in Switzerland. Unlike in previous studies, the main cause of mortality was infectious diseases (60.5%), followed by traumatic insults (27.7%). The lungs were the main organ affected, showing mostly lymphoplasmacytic (45.9%), granulomatous (18.1%) or suppurative (8.2%) pneumonia. Nematodes were detected in 57.2% of all lungs and were significantly associated with pneumonia (*p*-value < 0.001). To our knowledge, this is the first study to report infectious diseases as the main cause of hedgehog death, emphasizing the need for wildlife rescue centers to adopt appropriate diagnostic and therapeutic measures. Further research is necessary to determine the broad range of infectious agents that affect this species and elucidate their interplay with the host. Finally, citizen sensitization should be implemented to promote responsible behaviors that could reduce human-related traumatic events.

## 1. Introduction

The European hedgehog (*Erinaceus europaeus*; Linnaeus, 1758) is a small nocturnal mammal [1] considered endemic in most parts of Europe [2]. It inhabits lowlands and hilly areas rich in grass, trees and fallen leaves [3,4,5].

Although classified as insectivores [6], with a diet mainly consisting of invertebrates (e.g., earthworms and slugs), hedgehogs have also been reported to feed on small vertebrates, bird eggs, fruits, plants and pet food found in urban gardens [3,6,7,8].

In winter, they enter hibernation [9], usually beginning in October–November and ending in March–April [9,10,11]. Hibernation length and climatic conditions also influence hedgehog reproduction. Hedgehogs begin mating in spring, soon after arousing from hibernation [9]. During the mating period, males expand their home range, leading to an increased risk of traumatic injuries, particularly from road accidents [7,10,12]. Females usually have one litter per year in colder climates, delivering four to five hoglets at the beginning of the summer [9,12]. However, in milder climates and when abundant food resources are available, or if their first litter is lost, female hedgehogs can mate again and deliver a second litter from August onwards [11]. Nonetheless, this can pose a risk for both adult females and hoglets, as they are consequently more dependent on acquiring adequate food resources to increase fat deposits and survive winter [13].

Hedgehogs exhibit synanthropic behavior [14], meaning that they reach higher densities in urban areas than in rural ones [5,15,16]. In anthropogenic environments, they can find more shelter possibilities, an abundance of both prey and pet food, a warmer microclimate and fewer wild predators (especially badgers) [5,13,15,17,18,19]. However, urban areas pose a number of risks to hedgehogs, including habitat fragmentation with a consequent reduction in their genetic diversity [5,16,18,20], an increased risk of road accidents, predation by domestic animals, trauma [5,19,21,22,23] and pesticide intoxication [24,25,26].

Over the last few decades, several authors have reported a gradual decrease in hedgehog populations in several European countries (e.g., United Kingdom, the Netherlands, Sweden and Switzerland) [16,17,18,27]. The reasons underlying this phenomenon appear to be multifactorial, with human activities proposed to play a major role, along with climate change, predation and infectious diseases [13,16,19]. In Zurich (Switzerland), hedgehog populations have been estimated to have declined by 40.6% over 25 years, accompanied by a 17.6% reduction in their distribution [16]. On the other hand, hedgehog populations in Italy are considered stable and are classified as Least Concern in the IUCN (International Union for Conservation of Nature) Red List of Italian Vertebrates [28]. However, it should be noted that, to our knowledge, quantitative data on Italian hedgehog populations are not available [29].

The causes of hedgehog mortality have been the topic of several studies, but the vast majority has focused on anthropogenic causes, especially road deaths [6,10,22,30,31,32], or analyzed admission and survival rates from wildlife rescue centers (WRCs) [11,13,33]. Only a few authors have taken data resulting from postmortem examinations into account, linking the cause of death with the main pathological findings, but nonetheless, obtaining different results [19,23]. These discrepancies in the main causes of mortality and the overall scarcity of studies reporting pathological findings in hedgehogs, along with the declining populations across Europe, prompted us to initiate this study.

The study aimed to describe the main causes of admission, mortality and pathological findings observed in European hedgehogs submitted for full post-mortem examination, including histological investigation.

## 2. Materials and Methods

This study was conducted at the Unit of Veterinary Pathology of the Department of Veterinary Sciences, University of Turin (Italy), and at the Institute for Veterinary Pathology of Vetsuisse Faculty, University of Zurich (Switzerland). The distance, in a beeline, between the two regions, calculated considering the two main cities (Turin and Zurich) as starting and ending points, is roughly 270 km [34].

### 2.1. Animal Origin

The Italian Unit of Veterinary Pathology analyzed wild European hedgehogs, dead or euthanized, for severe pathological conditions from 2018 to 2022 in two WRCs of the Piedmont region, North-Western Italy: Centro Animali Non Convenzionali (C.A.N.C.), a section of the Veterinary Teaching Hospital of Turin University specialized in the rescue of all wildlife species, located in Grugliasco (Turin province), and Centro Recupero Ricci La Ninna (La Ninna), a private WRC involved in the hospitalization and rehabilitation of hedgehogs, sited in Novello (Cuneo province). All these animals were found in Piedmont region by citizens or members of law enforcement agencies and brought to both centers because they were considered in need of medical assistance.

The Swiss Institute of Veterinary Pathology of Zurich examined all hedgehogs submitted for full post-mortem evaluation by private owners, WRCs or veterinary clinics. The animals were either found dead or died/were euthanized in animal facilities between 2011 and 2022.

### 2.2. Study Animals

For each animal, origin, sex, age (determined according to morphological characteristics [7,9,19] and divided into three categories—unweaned, juvenile and adult), body condition score (BCS, recorded on a scale of 1 to 5, with 1 being cachectic and 5 obese), the reason for collection from its habitat or the clinical signs displayed at the WRCs/veterinary clinics before death (if available) and the date of death were recorded.

According to previous publications, the reasons for admission were classified as follows: poor overall condition (apathy, weakness etc.), trauma, respiratory signs, gastrointestinal signs, neurological signs, neoplasia, random find/orphan, found dead and unknown [19,23]. The category “random find” includes all animals unnecessarily brought to WRCs by people from the public [13,19,33].

Animals were euthanized due to clinical deterioration and poor prognosis. In Italy, euthanasia was performed by induction of general anesthesia and intracardiac inoculation of a medicinal preparation consisting of mebezonium iodide, embutramide and tetracaine. In Switzerland, all submitted animals had either naturally deceased or had already been euthanized at WRCs or local veterinary clinics. Information on the exact method of euthanasia was not provided in the Swiss cases.

### 2.3. Pathological Examination

All animals underwent a full post-mortem examination. Italian hedgehogs were examined by PhD students in Veterinary Sciences overseen by a full professor in Veterinary Pathology. In Switzerland, all necropsies were performed by residents of the European College of Veterinary Pathologists (ECVP) supervised by ECVP diplomates. The main tissues (liver, spleen, intestine, kidneys, lungs, heart and brain) and additional organs showing gross alterations were collected and fixed in 10% buffered formalin for at least 24 h, embedded in paraffin blocks, sectioned at 3 μm, and routinely stained with hematoxylin and eosin (HE).

The most likely cause of death was established on the basis of both macroscopic and histological findings. Five main categories were determined: trauma/predation, infectious disease, starvation, neoplasia and unknown [19,23].

In Swiss reports, information on the cause of death was extracted from the “Diagnoses” and “Commentary” sections of necropsy reports.

### 2.4. Bacteriological and Parasitological Investigations

In selected cases, and in particular, in animals in which the organ lesions raised the suspicion of an infectious process, additional examinations were initiated. These included routine bacteriological investigations (for anaerobic and/or aerobic agents, depending on the submitted organ sample) in cases in which there was evidence of suppurative/fibrinous inflammation. When a bacterial species with known pathogenic potential was isolated and it was compatible with the macroscopic/histological lesions detected, it was considered the responsible agent of pathology.

In animals that were severely emaciated/cachectic and/or displayed evidence of gastroenteritis, parasites in the gastrointestinal tract or a verminous pneumonia, fecal samples were collected, submitted and examined via standard sedimentation and flotation protocols to identify parasitic structures. Due to cost limitations, only a subset of animals (n = 96 for parasitological and n = 17 for bacteriological examination) was subjected to further laboratory analyses.

### 2.5. Statistical Analysis and Data Presentation

Statistical analysis was performed using IBM SPSS Statistics software (version 29.0.1.0 (171)). Descriptive data are provided as frequencies. Given the qualitative nature of the variables, chi-squared test and Fisher’s exact test (in case of two categories in both variables) were used to determine the possible presence of statistically significant differences between variables. The significance level was set at an alpha-value < 0.05.

In each Results subsection, descriptive statistics of collective data gathered from both countries and, if possible, associations between the different parameters are provided. In the subsection (Section 3.4), since most of the pathologic alterations affected only a small number of animals, the descriptive statistics on the pathological findings are reported collectively for all animals in both countries. In the last subsection (Section 3.4.5), because the examinations were performed only on a limited subset of animals, only descriptive statistics were provided and the country of origin was not considered.

## 3. Results

### 3.1. Descriptive Analysis

A total of 271 animals were included in the study. Of them, 162 animals (59.8%) originated from Italy and 109 animals (40.2%) from Switzerland. More specifically, with regards to Italian hedgehogs, 140 (51.7%) originated from C.A.N.C. and 22 animals (8.1%) from La Ninna WRC.

An overview of all population characteristics (sex, age and BCS) for each country of origin is provided in Table 1.

Table 2, Table 3 and Table 4 display the frequencies and percentages of hedgehog causes of admission and death, divided by the country of origin (Table 2), biological characteristics (sex, age and BCS; Table 3) and the season of mortality (Table 4).

Regarding the season of death, most hedgehogs died in summer (from 22nd June to 22nd September; n = 100, 36.9%), followed by spring (from 21st March to 21st June; n = 72, 26.6%), autumn (from 23rd September to 21st December; n = 66, 24.4%) and winter (from 22nd December to 20th March; n = 26, 9.6%). These data were not available for 7 animals (2.6%).

### 3.2. Causes of Admission

The main cause of hedgehog admission at veterinary clinics and WRCs of both countries was traumatic insults (n = 77, 28.4%, 95% CI: 23.1–34.2), followed by a poor overall condition (n = 75, 27.7%, 95% CI: 22.4–33.4), respiratory (n = 28, 10.3%, 95% CI: 7.0–14.6), gastrointestinal (n = 14, 5.2%, 95% CI: 2.9–8.5) or neurological signs (n = 6, 2.2%, 95% CI: 0.8–4.8) and random find without manifestation of specific clinical signs (n = 23, 8.5%, 95% CI: 5.5–12.5). A total of 30 animals were found dead or died prior to arrival at the centers (11.1%, 95% CI: 7.6–15.4), and for 16 individuals, the cause of admission was not recorded (5.9%, 95% CI: 3.4–9.4).

A significant association was observed between the cause of admission and the country of origin (*p*-value < 0.001, Cramer’s V = 0.593). As shown in Table 2, in Italy, most animals were admitted due to a traumatic injury (37.0%) or debilitation (34.6%), while in Switzerland, most animals were found dead (24.8%), followed by manifestations of weakness (17.4%) and traumatic insults (15.6%).

Moreover, considering both countries together, the causes of admission were significantly associated with age classes (*p*-value < 0.001, Cramer’s V = 0.298). Adult was the category with most animals admitted because of trauma (36.4%) or respiratory signs (14.3%) (see Table 3). In contrast, juveniles were mainly submitted due to the manifestation of weakness/debilitation (35.7%) or because they were found dead (or had died prior to therapeutic treatment; 14.3%). Unweaned animals were mainly randomly found (40%).

BCS also showed a significant association with the causes of admission (*p*-value = 0.002, Cramer’s V = 0.240). Emaciated or thin animals were mainly admitted due to clinical manifestation of weakness (n = 38, 32.5%, 95% CI: 24.1–41.8), while animals with a good or excellent body condition mainly were submitted due to trauma (n = 49, 31.8%, 95% CI: 24.6–39.8).

No significant association was found between the sex and the cause of admission.

### 3.3. Causes of Death

The summary of the causes of death for both countries revealed that most hedgehogs died of infectious or parasitic diseases (n = 164, 60.5%, 95% CI: 54.4–66.4), followed by traumatic insults (n = 75, 27.7%, 95% CI: 22.4–33.4). Five animals out of 271 (1.8%, 95% CI: 0.6–4.3) died of starvation (determined due to severe cachexia and emptiness of the gastrointestinal tract) and another five (1.8%, 95% CI: 0.6–4.3) because of neoplasia. For 22 animals, the macroscopic and histologic analysis could not determine the cause of death with certainty (8.1%). Euthanasia was performed on 63 animals (23.2%).

A statistically significant association between the cause of death and the country of origin was detected (*p*-value < 0.001, Cramer’s V = 0.325). In both countries infectious diseases were the most frequently reported cause of mortality (53.7% in Italy and 70.6% in Switzerland). On the other hand, as displayed in Table 2, a higher percentage of animals had died due to traumatic insults in Italy (38.3%), while only 13 animals were reported with trauma in Switzerland (11.9%).

Considering both countries together, a significant association was also present between the cause of mortality and BCS (*p*-value < 0.001, Cramer’s V = 0.253). Infectious/parasitic disease was the main cause of mortality across all body categories (see Table 3). However, animals with a good or excellent body condition showed a higher percentage of traumatic insults (34.5% and 33.3%, respectively).

Age and season of death also appeared to be statistically related (*p*-value < 0.001, Cramer’s V = 0.270), with adult animals mainly dying in spring (n = 58, 37.7%) and summer (n = 41, 26.6%), and juveniles predominantly dying in summer (n = 54, 48.2%) and autumn (n = 36, 32.1%). Even if a statistical association was not found (*p*-value = 0.498), there was a trend for male hedgehogs to most frequently decease in summer (n = 47, 33.3%) and spring (n = 42, 29.8%), while females mainly had died in summer (n = 45, 40.2%) and autumn (n = 31, 27.7%).

No association was either present between the cause of death and the sex (*p*-value = 0.952), age (*p*-value = 0.103) or season (*p*-value = 0.130).

### 3.4. Pathological Findings

Histologic examination was performed on 257 hedgehogs (94.8%) and 14 animals (5.2%) were excluded due to either severe autolytic processes or because the cause of death could be clearly stated based on gross examination. The data reported refer to all animals analyzed, considering comprehensively both countries.

#### 3.4.1. External Examination

Upon external gross examination, 65 animals (24.0%, 95% CI: 19.0–29.5) presented with ectoparasitic infestation. Ticks belonged to *Ixodes hexagonus* and *Ixodes ricinus*, while fleas were all *Archaeopsilla erinacei*, identified by means of taxonomic keys as part of a further research project [34,35,36]. The percentages of ectoparasite detection are reported in Table 5.

#### 3.4.2. Inner Organs

The descriptions of the macroscopic and histologic lesions detected in the inner organs are reported in Table 5 and Table 6 and are arranged based on the frequency in which each organ system was affected, ranging from most to less frequently interested. Figure 1 and Figure 2 represent some of the macroscopic and histologic lesions detected, respectively.

The lung was the most affected organ. In 57.2% of hedgehogs, cut sections of nematode eggs, larvae or adults were detected in the bronchial lumina (Figure 1e and Figure 2a). A significant association was found between parasite presence and histologic findings (*p*-value < 0.001, Cramer’s V = 0.525): in 91.8% (95% CI: 80.4–97.7) of the granulomatous pneumonia (n = 45), and in 65.3% (95% CI: 55.9–73.8) of the lymphoplasmacytic ones (n = 77), a concurrent presence of parasites was detected. In eight animals, the parasite presence was associated with lymphoplasmacytic laryngitis or tracheitis.

Cut sections of nematodes were detected in the intestinal lumen and/or in the mucosa of 35 animals (Figure 2d). A statistically significant association was observed between the presence of intestinal parasites and enteritis (*p*-value = 0.022, Cramer’s V = 0.180).

Of the 74 animals affected by skin lesions, chronic, suppurative or ulcerative dermatitis was macroscopically reported in 17 animals (23.0%, 95% CI: 14.0–34.2) and subjected to histological examination. Of those animals, three cases presented multifocal epidermal ulcerations with degenerated neutrophils, bacterial colonies and mixed cellular infiltration in the subcutis. Three further animals showed chronic dermatitis with orthokeratotic hyperkeratosis, of which, two displayed intradermal presence of fungal spores and hyphae.

#### 3.4.3. Body Cavities

The macroscopic findings detected in the body cavities are shown in Table 5 and Figure 1. Herniation of the abdominal organs was seen in four animals: three with lacerations of the abdominal wall with intestinal herniation and one with diaphragmatic intestinal herniation.

#### 3.4.4. Neoplasia

Neoplasia was overall not common and led to the death of five animals (1.8%). Two hedgehogs were affected by lymphoma (Figure 1i and Figure 2i); multifocally, the liver and spleen (and in one animal also the kidneys) were infiltrated by sheets of monomorphic round cells of 10–20 μm in diameter. Cells showed centrally located, heterochromatin-rich, round- to beam-shaped nuclei and sparse eosinophilic cytoplasm. Zero to one mitotic figure were detected in 10 high-power fields at highest magnification (400×).

One animal displayed an anaplastic tracheal carcinoma, characterized by a mucosal infiltration of small groups of pleomorphic neoplastic cells (Figure 2h). The polygonal cells possessed a moderate to abundant vacuolized cytoplasm, undefined margins, irregular nuclei with vesicular chromatin and a small nucleolus, with moderate to abundant anisocytosis and anisokaryosis and rare mitotic figures. Neutrophilic infiltration and necrotic foci were also present multifocally in the neoplasm.

One hedgehog showed multiple white, smooth, nodular proliferations on the thoracic wall, the abdominal serosal face of the diaphragm, kidneys and spleen. Histologically, the neoplasia showed features of a sarcoma, as it consisted of moderately pleomorphic elongated, spindeloid cells forming whorls and streams. Due to the combination of the macroscopic and histologic appearance of the neoplasia, the tentative diagnosis of a mesothelioma was made.

Another hedgehog was diagnosed with a sarcoma, that was presented as a nodule of 6 cm × 6 cm × 4 cm found in the subcutis of the left lateral abdominal region. Histologically, the proliferation was composed of spindle cells arranged in irregular streams and bundles with a variable amount of eosinophilic fibrillar cytoplasm, elongated to round nuclei and inconspicuous nucleoli. Rare mitotic figures were observed. Metastases of the neoplastic proliferations were found in the omentum majus, the liver, the spleen, the kidney, the lungs and the right submandibular lymph node.

#### 3.4.5. Detection of Parasitic and Bacterial Infectious Agents

In 96 animals, fecal samples were submitted to coprological examination. From those, endoparasites could be detected in 77 samples (80.2%, 95% CI: 70.8–87.6). More specifically, 68 (88.3%, 95% CI: 79.0–94.5) animals were infested by *Capillaria* spp. (in most cases these could be further identified as the species *Capillaria aerophila*), 37 (48.1%, 95% CI: 36.5–59.7) by *Crenosoma striatum*, 13 (16.9%, 95% CI: 9.3–27.1) by *Brachylaemus erinacei* and six (7.8%, 95% CI: 2.9–16.2) by *Cystoisospora* spp. Of all positive animals, 42 animals (54.5%, 95% CI: 42.8–65.9) displayed coinfection with at least two or all three of the addressed parasite species.

Bacteriological investigations were performed on 17 animals. *Clostridium perfringens*, *Enterococcus* spp. or *Enterobacteriaceae* were isolated from the gastrointestinal tract in three cases. In the respiratory tract, *Pasteurella* spp. and *Pasteurella multocida*, *Bacteroides fragilis*, *Salmonella enterica* serovar *enteritidis* or *Morganella morganii* were isolated from four animals. One hedgehog with multiple abscesses and peritonitis was positive for *Clostridium perfringens* type A. In nine further samples, no predominant bacterial agent could be cultivated, as only either a mixed bacterial flora or no bacteria (negative result) were reported.

## 4. Discussion

This study aimed to evaluate the causes of admission and mortality in European hedgehogs and to describe their main pathologic findings by means of a full postmortem examination, including histological investigation. This species is distributed throughout Europe and, as its presence is tightly connected to human settlements, it plays a crucial role in the human—wildlife interface. Recent studies have described a decline in the European hedgehog population in several European areas [16,17,18,27]. This prompted us to initiate the present study, by collecting and comparing data from two neighboring countries, Italy and Switzerland.

The Italian Piedmont region and Swiss Canton of Zurich are characterized by similar geographical features. Both regions are located in the subcontinental climatic European area, presenting mild summers and cold winters and seasonal fluctuations in temperature extremes [37]. Both countries are rich in rivers and lakes, while forest coverage accounts for approximately 30% in the Canton of Zurich and 37% in Piedmont [38,39]. Moreover, both regions are highly inhabited, with populated valleys characterized by marked land use and industrial development [38,40]. Climate change, as a global phenomenon, also affects both areas, and manifests as glacier melting, high-temperature peaks, reduction in rainfall and an increase in both intense rain and drought periods [41,42]. These similar characteristics allowed us to reduce the possible bias represented by the different geographic origins of the sampled animals and to facilitate the comparison of the obtained results.

In general, studies have pointed out that both climate change and human activities may contribute to a decline in hedgehog population [16,18]. More specifically, a recent study by Taucher indicated a decline in the population of European hedgehogs in the Canton of Zurich, which, among many other causes, was attributed to urban densification with consequent habitat fragmentation, changes in summer and winter temperatures, reductions in insect biomass, pesticide use and infectious diseases [16]. In addition, considering the geographic similarities between the two countries [29] and the lack of studies on the causes of mortality in Italian hedgehog populations, we considered this analysis fundamental, especially in the eye of possible future decline in population density.

In our study, the main causes of admission of the animals in both countries were traumatic insults, followed by debilitation, finding of a dead animal or death during transport, respiratory signs, and random findings. However, the frequencies of the causes of admission significantly differed between the two countries, with Italy constituting mainly traumatic insults (37.0%) and debilitation (34.6%), whereas in Switzerland, most of the examined hedgehogs were either found dead (24.8%) or displayed weakness (17.4%). This difference could be attributed to a possible bias related to the difference in the sequence of procedures preceding the examination of the carcasses: Italian hedgehogs were first submitted to the WRCs and—in the case of death or euthanasia—subsequently submitted for postmortem examination, while in Switzerland, all animals were directly submitted for necropsy by citizens, veterinary clinics or WRCs. Therefore, it could be less likely for Italian citizens to bring animals to WRCs, when they are found dead or have already died during transport. In Italy, 38.3% of animals displayed lethal trauma, while this cause of death only accounted for 11.9% of examined hedgehogs in Switzerland. Although this information was not included in the statistical analysis, in most of the examined cases, the anamnesis indicated that trauma was most often related to car accidents, predation, entanglement or mowing-provoked lacerations. The difference in the frequency of trauma between the two countries could be attributed to differences in the urban environments of Italy and Switzerland. Indeed, areas with limited driving speed have been introduced in Zurich and in many other Swiss cities, where only a slight increase in the number of vehicles has been recorded [16]. In addition, Swiss citizens use public transport more frequently than in Italy, particularly in the Piedmont region (76.7% vs. 16.2%) [43,44]. These factors most likely contributed to the reduction in possible road traffic accidents. Moreover, Switzerland has a higher population density (215 inhabitants/km^2^ on average, reaching over 400 inhabitants/km^2^ in the Central Plateau, a geographical area where most of the population is located) [45] compared to the Piedmont region (168.4 inhabitants/km^2^ and 325.1 inhabitants/km^2^ in Turin province) [46]. This urban densification also led to a reduction in urban green spaces in Switzerland [16], limiting traumatic insults related to gardening-related activities (e.g., entanglement, mowing wounds, predation by domestic animals as dogs). Finally, the Swiss government has focused on promoting campaigns that help implement hedgehog-friendly gardens and educate citizens regarding responsible behavior towards this species, reducing the likelihood of human-related traumatic insults [47].

The causes of admission and death were also significantly associated with BCS. Traumas mainly affected animals with good or excellent body condition, which are more active than emaciated individuals and therefore cover larger areas, which simultaneously imposes a greater risk of being trapped, hit by cars or predated [7].

Traumatic insults mainly affected adults (65.3%) and were more frequent in the spring (37.3%) and summer (36.0%). It is important to note that male hedgehogs expand their territory in spring to mate, and that females are more active in summer, as they look for food resources (so as to increase their fat tissue deposition after or during lactation). In conclusion, our study supports the theory already stated by other authors, that increased locomotion related to the reproductive season can increase the probability of traumatic insults [6,7,10,12,13,32].

Infectious diseases mainly occurred in summer (36.6%) and autumn (27.4%). In these seasons, adults show reduced energy levels after the mating season (mainly males) or due to increased energy consumption during pregnancy and lactation (for female hedgehogs, which can deliver up to two litters in those seasons) [32]. Reductions in body conditions are known to predispose animals to infections, starvation and dehydration [32]. In addition, summer and autumn also represent the seasons in which juvenile hedgehogs start to disperse and search for food resources to build up adequate fat reserves prior to hibernation. The exploration of new territories also explains the high percentage of random finds of juveniles (16.1%) in our study. Indeed, citizens often wrongly believe that young animals found in the wild are abandoned or in danger and, therefore, needlessly submit them to WRCs. However, this can reduce their chances of survival [9,13].

The diagnosis of infectious disease was based on macroscopic and/or histologic detection of inflammatory infiltrates in at least one organ. In 39% of the cases in which an infection was diagnosed, the presence of leukocytes in the vessels of different organs, accompanied or not by the presence of fibrinous thrombi (indication of disseminated intravascular coagulation) and bacteria, justified the suspicion of terminal sepsis (Figure 2b). To our knowledge, this study is the first to report infectious diseases as the most frequent cause of mortality in European hedgehogs. In two other studies focusing on the results of full postmortem examinations in this species, traumatic insults of known or unknown origin were determined as the most frequent cause of mortality, leading to death in 41% of animals in the study by Zacharopoulou et al. [23], and in 32.7% in the study by Gracês et al. [19]. The first study determined infectious diseases caused by viruses or bacteria as the second most frequent cause of mortality (34%), followed by parasitism (11%), whereas infectious agents were responsible for 8.1% of hedgehog deaths in the study by Garcês et al. However, if we consider infectious and parasitic diseases together in Zacharopoulou’s work, the overall prevalence is 45%, which is higher than the frequency of traumatic insults, and, therefore, would also constitute the first cause of mortality. In the study by Garcês et al. [19], the authors stated that histopathological examination was frequently impaired by severe autolysis or was not performed because of a lack of financial resources, thus possibly preventing the detection of lesions compatible with an infectious origin.

In all the addressed studies, verminous pneumonia (infection with lungworms) represented one of the most described infectious diseases in the European hedgehog [16,23,48]. In our study, the lungs were the most affected organ. The majority of animals showed lymphoplasmacytic pneumonia, followed by granulomatous and suppurative pneumonia. All types of pneumonia were significantly associated with the histological presence of parasites. This is in line with the results of fecal exams, which identified the presence of the nematodes *Crenosoma striatum* and *Capillaria aerophila*, both of which have already been described as the main agents of verminous pneumonia in hedgehogs [48]. Infection with both parasites has been associated with clinical signs such as weight loss, dry cough, nasal discharge, dyspnoea, wheezing and exercise intolerance [49]. Histologically, adult *Capillaria aerophila* nematodes are found either free in the airway lumen and/or within the airway epithelium, from the larynx to the bronchioles, as stated in a recent study on European hedgehogs originating from the Canton of Zurich. In this study, embedded worms and eggs of *Capillaria aerophila* were associated with epithelial hyperplasia or metaplasia of the respiratory epithelium [48]. In contrast, *Crenosoma striatum* adults were predominantly free in the lumen of bronchi and bronchioles, and their larvae were occasionally seen in granulomas in the pulmonary interstitium [48]. In accordance with previous studies, we also identified co-infection with these two parasite species [48,49,50,51,52,53,54]. Lungworm infection has also been reported to occur more frequently in adults than in juveniles [49,51,53], which is thought to be related to the shorter exposure time of young hedgehogs to the parasite’s intermediate or paratenic hosts (e.g., earthworms and snails). This difference in age susceptibility also explains why the hedgehogs admitted due to the manifestation of respiratory clinical signs in our study were mainly adults (14.3%, compared to 5.4% juveniles).

The clinical relevance and significance of lungworm infection in hedgehogs have been controversial in previous studies. Although some authors have stated that lungworm infection is a main driver of hedgehog mortality [7,49,55], others have also associated parasites with subclinical infection/disease, indicating that a certain parasitic burden might represent a part of the hedgehog ecology [23,48]. In support of this hypothesis, Zacharopoulou et al. [23] reported that parasitism was associated with mortality in only 11% of animals. In accordance with the occasionally contradictory conclusions of previous studies, the interpretation of the clinical significance (and association with mortality) of lungworm infection was not always straightforward in the present study. Although a high number of deceased or euthanized animals displayed evidence of lungworm infection and a certain degree of associated pneumonia, the affected animals showed variation in their overall health condition. Hence, verminous pneumonia was detected in otherwise healthy animals (e.g., ones that had died due to a traumatic insult, without other histologic lesions in the inner organs) but also in hedgehogs displaying additional concurrent disease processes in other inner organs. In a small proportion of animals, verminous pneumonia was the only disease process noted on postmortem examination, highlighting its importance as a major contributor to mortality. Based on these results, we can hypothesize that, while in some cases, lungworm infection might not cause drastic alterations in host homeostasis, in other cases, it might represent the main agent of disease or promote the development and spread of other infectious agents, ultimately leading to the death of the animal. Nevertheless, the exact mechanism by which lungworms disrupt the host immune system or interact with other pathogens warrants further investigation.

Indeed, it has been reported that verminous pneumonia can be complicated by secondary bacterial infections (e.g., *Bordetella bronchiseptica* or *Pasteurella multocida*) that might subsequently lead hedgehogs to death [56]. This was also the case in two animals in our study, as *Pasteurella* spp. and *Pasteurella multocida* were isolated from the respiratory tract in addition to a lungworm infection being discovered. However, because of cost limitations, the majority of the lung samples were not subjected to bacteriological culture and it was not possible to test this hypothesis and to draw statistically significant conclusions on a possible lungworm—bacteria correlation. Therefore, further studies are needed to determine the effects of endoparasites and possible co-infections on the overall health of European hedgehogs.

Additionally, parasites detected in the enteric lumen displayed a statistically significant association with the presence of lymphoplasmacytic enteritis. European hedgehogs have been reported as hosts for several endoparasites [49,53]. Among these, *Capillaria* spp., *Physaloptera clausa* and *Brachylaemus erinacei* are the parasites most frequently found in the gastrointestinal tract, where they have been described in association with chronic non-suppurative gastroenteritis [49,51,54,57]. In our fecal exams, *Capillaria* spp. and *Brachylaemus erinacei* were detected, explaining the association between enteritis and parasite presence.

Infection with several bacterial pathogens has been described in European hedgehogs: *Bordetella bronchiseptica*, *Corynebacterium ulcerans*, *Escherichia coli, Klebsiella* spp., *Pasteurella* spp., *Pseudomonas* spp., *Staphylococcus* spp., *Streptococcus* spp. and *Yersinia* spp. have all been isolated from diseased hedgehogs [6,7,21]. In our study, bacterial isolation was performed on a limited number of samples (17). Most of these (nine) yielded mixed flora or no bacterial culture. In some samples (three), *Clostridium perfringens*, *Enterococcus* spp., or *Enterobacteriaceae* were isolated, but even though these species have been described as having zoonotic potential [58,59,60], we were not able to correlate them with the pathological lesions detected. In other animals (five), however, the bacterial isolation and the macroscopic and histological findings led us to conclude that the isolated species could be considered agents of disease. One animal affected by fibrinous pleuropneumonia tested positive for *Morganella morganii*. This opportunistic pathogen has been described in both animals and humans (with or without evidence of disease), and also has zoonotic potential [61]. In another hedgehog with a suspected septic process, *Salmonella enterica* serovar *enteritidis* was isolated from the lungs. *Salmonella* spp. was responsible for the death of two hedgehogs in Denmark [18] and *Salmonella enterica* serovar *enteritidis* was identified in systemic and pulmonary infections in European hedgehogs in Finland [6]. This particular serovar also has zoonotic potential and Lawson et al. have identified hedgehogs as sources of transmission of this pathogen to humans [62]. Hedgehogs, indeed, have been repeatedly reported as reservoirs for several zoonotic agents (e.g., MRSA *Staphylococcus aureus, Leptospira* spp. and several tick-borne pathogens) [21,63,64].

Overall, non-infectious lesions were uncommon in the examined hedgehogs, with 1.8% of all the animals displaying neoplasia. Similarly, Garcês and colleagues identified one case of neoplasia in 248 necropsied animals [19], whereas in Zacharopoulou’s study neoplasia was found in 6% of all animals (2 out of 35) [23]. Similar to other mammalian species, different types of tumors have been reported in European hedgehogs, including mammary tumors [19,65], cutaneous malignant peripheral nerve sheath tumor and cortical adrenal gland adenoma [23], desmoplastic ganglioglioma [66], subcutaneous myxosarcoma [67] and lacrimal duct carcinoma [68]. In conclusion, neoplastic diseases appear to be rare in this species.

## 5. Conclusions

The present study examined the main causes of mortality and pathologic findings of European hedgehogs in Italian Piedmont region and the Swiss Canton of Zurich, regions characterized by similar geographical features. Trauma, especially road accidents or predation by domestic or wild animals, is the most common cause for submission of this species to WRCs and a significant reason for mortality. Therefore, raising public awareness on how to effectively reduce interactions and successfully protect this species’ habitat is of fundamental importance to reduce its mortality. Disseminating guidelines or educational campaigns could promote responsible behavior among citizens, providing information on how to create hedgehog-friendly gardens and the correct actions to take to reduce human-related traumatic events.

We also highlight the importance of infectious, and particularly parasitic, diseases contributing to hedgehog mortality, which should be considered when establishing an appropriate therapeutic approach. However, further microbiological studies are necessary to determine the main infectious agents found in this species. This is crucial not only for the overall health of this species but also considering the synanthropic behavior of hedgehogs and their contact with domestic animals and humans, to whom these pathogens could be transmitted. The role of endoparasites and their interaction with other infectious agents also warrants further investigation before establishing prophylactic measures. In light of the numerous reports indicating a decline in hedgehog populations in several European regions, frequent monitoring of the population health status is strongly needed in all countries.

## Figures and Tables

**Figure 1 animals-14-01852-f001:**
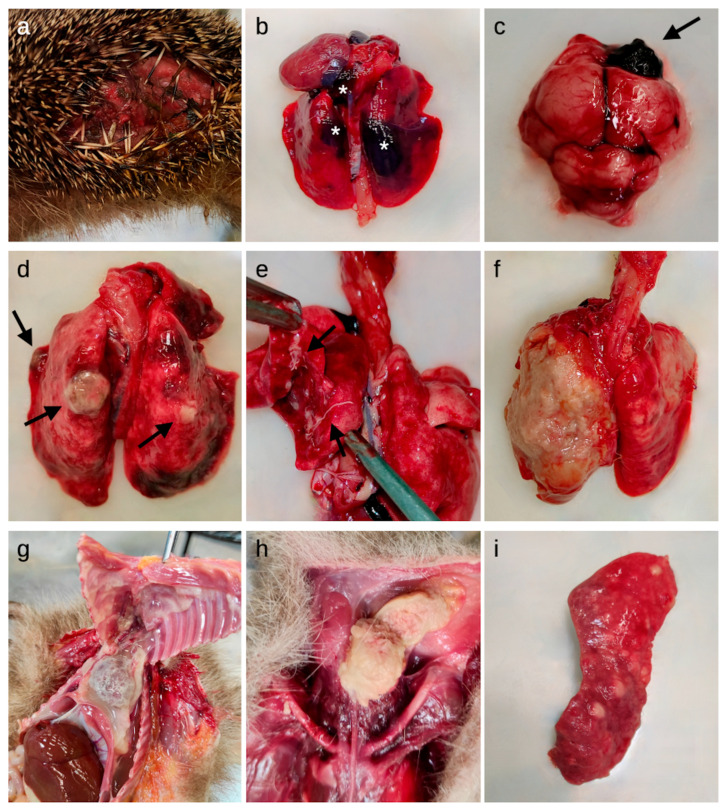
Macroscopic lesions detected in the studied European hedgehogs. (**a**) Skin: laceration of the lateral abdominal region; (**b**) Lungs: multifocal hemorrhages (asterisks) in the parenchyma; (**c**) Brain: focal hemorrhage in the right frontal lobe (arrow); (**d**) Lungs: multifocal granulomatous lesions (arrows) and hyperemia; (**e**) Lungs: lungworms (arrows) on the cut surface and in the bronchial lumina; (**f**) Lungs: severe diffuse fibrinous-suppurative pleuropneumonia; (**g**) Thoracic cavity: severe diffuse fibrinous pericarditis, pleuritis and serum-fibrinous thoracic effusion; (**h**) Ventral neck region: abscess; (**i**) Spleen: lymphoma.

**Figure 2 animals-14-01852-f002:**
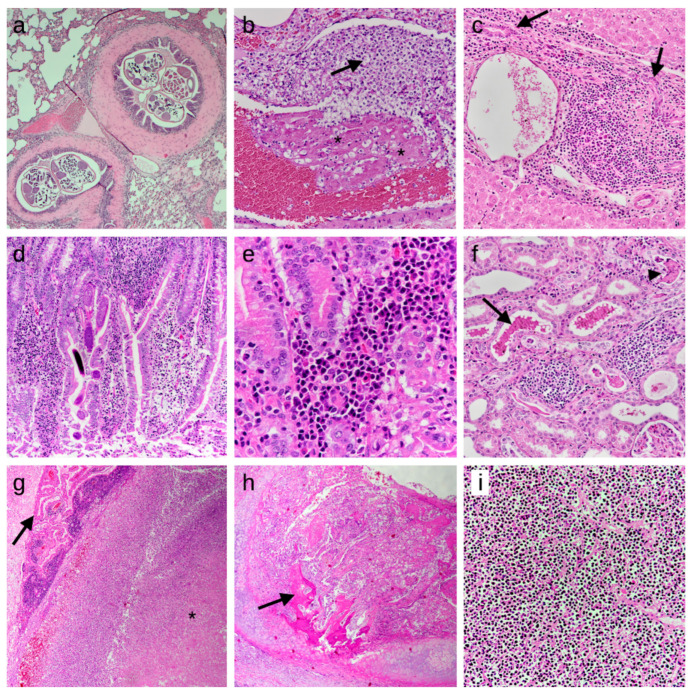
Histologic findings detected in the studied European hedgehogs. Lesions are shown according to the frequency of detection and the order in which they appear in the text. The first seven images represent inflammatory processes, while the last two are from neoplastic cases. HE stain. (**a**) Lung: lymphoplasmacytic bronchopneumonia with nematode slices in bronchial lumina, 50×; (**b**) Lung: presence of fibrin thrombi (asterisks) and immature neutrophils (arrow) attached to a pulmonary vessel wall, indicating septicemia/disseminated intravascular coagulation, 200×; (**c**) Liver: moderate chronic periportal hepatitis, characterized by infiltrations of lymphocytes and plasma cells and bile duct hyperplasia (arrows), 200×; (**d**) Intestine: moderate chronic enteritis with multiple larvae of nematodes in the mucosa, 100×; (**e**) Intestine: lymphocytes and plasma cells in the tela submucosae, 400×; (**f**) Kidney: mild chronic interstitial nephritis, characterized by infiltrations of lymphocytes and plasma cells. Also note the presence of chromoproteinuria (arrow) and cellular casts (arrowhead) in the tubular lumen, 200×; (**g**) Brain: focal abscess in the pons region (asterisk), compressing the choroid plexus (arrow), 50×; (**h**) Trachea: tracheal carcinoma, focally infiltrating the tracheal cartilage and showing focal osteoid formation (arrow), 25×; (**i**) Spleen: lymphoma, characterized by the presence of lymphoid neoplastic cells, entirely effacing the splenic architecture, 200×.

**Table 1 animals-14-01852-t001:** Population characteristics of the European hedgehogs included in the study, shown for both countries of origin. For Body Condition Score (BCS): Cachectic = 1/5; Decent = 2/5; Good = 3/5; Excellent = 4/5. No animals were scored 5/5 (obese); therefore, this category is not presented in the table.

		Sex						Age						BCS							
		Male		Female		Unknown		Adult		Juvenile		Unweaned		Cachectic		Decent		Good		Excellent	
		n	%, [95% CI]	n	%, [95% CI]	n	%, [95% CI]	n	%, [95% CI]	n	%, [95% CI]	n	%, [95% CI]	n	%, [95% CI]	n	%, [95% CI]	n	%, [95% CI]	n	%, [95% CI]
Country of origin	Italy	87	61.7, [53.1–69.8]	75	66.4, [56.9–75.0]	0	0.0, [0.0–19.5]	84	54.5, [46.3–62.6]	76	67.9, [58.3–76.4]	2	40.0, [5.2–85.3]	10	28.6, [14.6–46.3]	50	61.0, [49.6–71.6]	98	66.2, [58.0–73.8]	4	66.7, [22.3–95.7]
	Switzerland	54	38.3, [30.2–46.9]	38	33.6, [25.0–43.1]	17	100.0, [80.5–100.0]	70	45.5, [37.4–53.7]	36	32.1, [23.6–41.6]	3	60.0, [14.7–94.7]	25	71.4, [53.7–85.4]	32	39.0, [28.4–50.4]	50	33.8, [26.2–42.0]	2	33.3, [43.3–77.7]

**Table 2 animals-14-01852-t002:** Frequencies and percentages of the causes of admission and death of all examined hedgehogs, shown for both countries of origin.

		Country of Origin			
		Italy		Switzerland	
		n	%, [95% CI]	n	%, [95% CI]
Cause of admission	Trauma/Predation	60	37.0, [29.6–45.0]	17	15.6, [9.4–23.8]
	Debilitation	56	34.6, [27.3–42.4]	19	17.4, [10.8–25.9]
	Found dead	3	1.9, [0.4–5.3]	27	24.8, [17.0–34.0]
	Respiratory signs	13	8.0, [4.3–13.3]	15	13.8, [7.9–21.7]
	Random find	23	14.2, [9.2–20.5]	0	0.0, [0.0–3.3]
	Gastrointestinal signs	2	1.2, [0.1–4.4]	12	11.0, [5.8–18.4]
	Neurological signs	0	0.0, [0.0–2.3]	6	5.5, [2.0–11.6]
	Neoplasia	2	1.2, [0.1–4.4]	0	0.0, [0.0–3.3]
	Unknown	3	1.9, [0.4–5.3]	13	11.9, [6.5–19.5]
Cause of death	Infectious disease	87	53.7, [45.7–61.6]	77	70.6, [61.2–79.0]
	Trauma/Predation	62	38.3, [30.8–46.2]	13	11.9, [6.5–19.5]
	Starvation	0	0.0, [0.0–2.3]	5	4.6, [1.5–10.4]
	Neoplasia	2	1.2, [0.1–4.4]	3	2.8, [0.6–7.8]
	Unknown	11	6.8, [3.4–11.8]	11	10.1, [5.1–17.3]

**Table 3 animals-14-01852-t003:** Frequencies and percentages of the causes of admission and death of all examined hedgehogs, divided according to their biological characteristics (sex, age and BCS).

		Sex						Age						BCS							
		Male		Female		Unknown		Adult		Juvenile		Unweaned		Cachectic		Decent		Good		Excellent	
		n	%, [95% CI]	n	%, [95% CI]	n	%, [95% CI]	n	%, [95% CI]	n	%, [95% CI]	n	%, [95% CI]	n	%, [95% CI]	n	%, [95% CI]	n	%, [95% CI]	n	%, [95% CI]
Cause of admission	Trauma/ Predation	43	31.9, [24.1–40.4]	32	30.5, [21.9–40.2]	2	11.8, [1.5–36.4]	56	36.4, [28.8–44.5]	21	18.8, [12.0–27.2]	0	0.0, [0.0–52.2]	6	17.1, [6.6–33.6]	22	26.8, [17.6–37.8]	47	31.8, [24.4–40.0]	2	33.3, [4.3–77.7]
	Debilitation	37	27.4, [20.1–35.7]	35	33.3, [24.4–43.2]	3	17.6, [3.8–43.4]	34	22.1, [15.8–29.5]	40	35.7, [26.9–45.3]	1	20.0, [0.5–71.6]	9	25.7, [12.5–43.3]	29	35.4, [25.1–46.7]	37	25.0, [18.3–32.8]	0	0.0, [0.0–45.9]
	Found dead	14	10.4, [5.8–16.8]	8	7.6, [3.3–14.5]	8	47.1, [23.0–72.2]	13	8.4, [4.6–14.0]	16	14.3, [8.4–22.2]	1	20.0, [0.5–71.6]	7	20.0, [8.4–36.9]	8	9.6, [4.3–18.3]	15	10.1, [5.8–16.2]	0	0.0, [0.0–45.9]
	Respiratory signs	16	11.9, [6.9–18.5]	12	11.4, [6.0–19.1]	0	0.0, [0.0–19.5]	22	14.3, [9.2–20.8]	6	5.4, [2.0–11.3]	0	0.0, [0.0–52.2]	5	14.3, [4.8–30.3]	6	7.3, [2.7–15.2]	15	10.1, [5.8–16.2]	2	33.3, [4.3–77.7]
	Random find	14	10.4, [5.8–16.8]	9	8.6, [4.0–15.6]	0	0.0, [0.0–19.5]	3	1.9, [0.4–5.6]	18	16.1, [9.8–24.2]	2	40.0, [5.3–85.3]	1	2.9, [0.1–14.9]	7	8.5, [3.5–16.8]	15	10.1, [5.8–16.2]	0	0.0, [0.0–45.9]
	Gastrointestinal signs	8	5.9, [2.6–11.3]	5	4.8, [1.6–10.8]	1	5.9, [0.1–28.7]	8	5.2, [2.3–10.0]	5	4.5, [1.5–10.1]	1	20.0, [0.5–71.6]	2	5.7, [0.7–19.6]	4	4.9, [1.3–12.0]	8	5.4, [2.4–10.4]	0	0.0, [0.0–45.9]
	Neurological signs	2	1.5, [0.2–5.2]	3	2.9, [0.6–8.1]	1	5.9, [0.1–28.7]	4	2.6, [0.7–6.5]	2	1.8, [0.2–6.3]	0	0.0, [0.0–52.2]	0	0.0, [0.0–10.0]	3	3.7, [0.8–10.3]	3	2.0, [0.4–5.8]	0	0.0, [0.0–45.9]
	Neoplasia	1	0.7, [0.0–4.1]	1	1.0, [0.0–5.2]	0	0.0, [0.0–19.5]	2	1.3, [0.2–4.6]	0	0.0, [0.0–3.2]	0	0.0, [0.0–52.2]	0	0.0, [0.0–10.0]	1	1.2, [0.0–6.6]	0	0.0, [0.0–2.5]	1	16.7, [0.4–64.1]
	Unknown	6	4.4, [1.6–9.4]	8	7.6, [3.3–14.5]	2	11.8, [1.5–36.4]	12	7.8, [4.1–13.2]	4	3.6, [1.0–8.9]	0	0.0, [0.0–52.2]	5	14.3, [4.8–30.3]	2	2.4, [0.3–8.5]	8	5.4, [2.4–10.4]	1	16.7, [0.4–64.1]
Cause of death	Infectious disease	86	61.0, [52.4–69.1]	66	58.4%, [48.8–67.6]	12	70.6, [44.0–89.7]	89	57.8, [49.6–65.7]	70	62.5, [52.9–71.5]	5	100.0, [47.8–100.0]	22	62.9, [44.9–78.5]	53	64.6, [53.3–74.9]	86	58.1, [49.7–66.2]	3	50.0, [11.8–88.2]
	Trauma/ Predation	39	27.7, [20.5–35.8]	32	28.3, [20.2–37.6]	4	23.5, [6.8–49.9]	49	31.8, [24.6–39.8]	26	23.2, [15.8–32.1]	0	0.0, [0.0–52.2]	3	8.6, [1.8–23.1]	19	23.2, [14.6–33.8]	51	34.5, [26.8–42.7]	2	33.3, [4.3–77.7]
	Starvation	2	1.4, [0.2–5.0]	3	2.7, [0.6–7.6]	0	0.0, [0.0–19.5]	3	1.9, [0.4–5.6]	2	1.8, [0.2–6.3]	0	0.0, [0.0–52.2]	5	14.3, [4.8–30.3]	0	0.0, [0.0–4.4]	0	0.0, [0.0–2.5]	0	0.0, [0.0–45.9]
	Neoplasia	3	2.1, [6.1]	2	1.8, [0.2–6.2]	0	0.0, [0.0–19.5]	5	3.2, [1.2–7.4]	0	0.0, [0.0–3.2]	0	0.0, [0.0–52.2]	1	2.9, [0.1–14.9]	1	1.2, [0.0–6.6]	2	1.4, [0.1–4.8]	1	16.7, [0.4–64.1]
	Unknown	11	7.8, [4.0–13.5]	10	8.8, [4.3–15.6]	1	5.9, [0.1–28.7]	8	5.2, [2.3–10.0]	14	12.5, [7.0–20.1]	0	0.0, [0.0–52.2]	4	11.4, [3.2–26.7]	9	11.0, [5.1–19.8]	9	6.1, [2.8–11.2]	0	0.0, [0.0–45.9]

**Table 4 animals-14-01852-t004:** Frequencies and percentages of the causes of death of all examined hedgehogs, shown for all four seasons. Winter = from 22nd December to 20th March; spring = from 21st March to 21st June; summer = from 22nd June to 22nd September; autumn = from 23rd September to 21st December.

		Cause of Death									
		Infectious Disease		Neoplasia		Starvation		Trauma/Predation		Unknown	
		n	%, [95% CI]	n	%, [95% CI]	n	%, [95% CI]	n	%, [95% CI]	n	%, [95% CI]
Season of death	Winter	19	11.6, [7.1–17.5]	1	20.0, [0.5–71.6]	1	20.0, [0.5–71.6]	2	2.7, [0.3–9.3]	3	13.6, [2.9–34.9]
	Spring	37	22.6, [16.4–29.7]	2	40.0, [5.3–85.3]	1	20.0, [0.5–71.6]	28	37.3, [26.4–49.3]	4	18.2, [5.2–40.3]
	Summer	60	36.6, [29.2–44.5]	0	0.0, [0.0–52.2]	3	60.0, [14.7–94.7]	27	36.0, [25.2–47.9]	10	45.5, [24.4–67.8]
	Autumn	45	27.4, [20.8–34.9]	2	40.0, [5.3–85.3]	0	0.0, [0.0–52.2]	14	18.7, [10.6–29.3]	5	22.7, [7.8–45.4]
	Unknown	3	1.8, [0.4–5.3]	0	0.0, [0.0–52.2]	0	0.0, [0.0–52.2]	4	5.3, [1.5–13.1]	0	0.0, [0.0–15.4]

**Table 5 animals-14-01852-t005:** Frequency and percentage of the main macroscopic findings identified in the inner organs.

Organ	Macroscopic Findings	n	%, [95% CI]
Ectoparasites	Co-infestation	25	9.2, [6.1–13.3]
	Myiasis	17	6.3, [3.7–9.9]
	Ticks	15	5.5, [3.1–9.0]
	Fleas	8	3.0, [1.3–5.7]
Lung	Granulomatous bronchopneumonia ± pleuritis	90	33.2, [27.6–39.2]
	Catarrhal bronchopneumonia ± pleuritis	55	20.3, [15.7–25.6]
	Hyperemia	26	9.6, [6.4–13-7]
	Suppurative bronchopneumonia ± pleuritis	14	5.2, [2.9–8.5]
	Hemorrhage	6	2.2, [0.8–4.8]
	Fibrinous pleuropneumonia	4	1.5, [0.4–3.7]
	Neoplasia	1	0.4, [0.0–2.0]
Liver	Hyperemia	11	4.1, [2.0–7.1]
	Lipidosis	11	4.1, [2.0–7.1]
	Chronic hepatitis/Perihepatitis	5	1.8, [0.6–4.3]
	Rupture	4	1.5, [0.4–3.7]
	Neoplasia	1	0.4, [0.0–2.0]
Skin	Laceration/Abrasion/Hemorrhage	48	17.7, [13.4–22.8]
	Dermatitis	17	6.3, [3.7–9.9]
	Abscess/Granuloma	9	3.3, [1.5–6.2]
Intestine	Catarrhal enteritis	20	7.4, [4.6–11.2]
	Herniation	4	1.5, [0.4–3.7]
	Hemorrhagic enteritis	3	1.1, [0.2, 3.2]
Mesenteric lymph nodes	Reactive hyperplasia	20	7.4, [4.6–11.2]
Musculo-skeletal	Limb fracture/Luxation/Amputation	15	5.5, [3.1–9.0]
	Skull fracture	12	4.4, [2.3–7.6]
	Myositis/Osteitis	9	3.3, [1.5–6.2]
	Hemorrhage	8	3.0, [1.3–5.7]
	Polytrauma	4	1.5, [0.4–3.7]
Spleen	Congestion	16	5.9, [3.4–9.4]
	Neoplasia	3	1.1, [0.2, 3.2]
	Splenitis	3	1.1, [0.2, 3.2]
	Rupture	2	0.7, [0.1–2.6]
Kidneys	Interstitial nephritis/Glomerulonephritis	3	1.1, [0.2, 3.2]
	Hyperemia	2	0.7, [0.1–2.6]
	Neoplasia	2	0.7, [0.1–2.6]
	Petechiae	2	0.7, [0.1–2.6]
Stomach	Catharral gastritis	15	5.5, [3.1–9.0]
	Hemorrhagic gastritis	11	4.1, [2.0–7.1]
	Gastric ulcer	5	1.8, [0.6–4.3]
	Traumatic rupture	1	0.4, [0.0–2.0]
	Parasites	1	0.4, [0.0–2.0]
Brain	Hemorrhage	9	3.3, [1.5–6.2]
	Subdural hematoma	8	3.0, [1.3–5.7]
	Meningitis/meningoencephalitis	1	0.4, [0.0–2.0]
Heart	Fibrinous pericarditis	6	2.2, [0.8–4.8]
Urinary bladder	Suspected neurogenic bladder dysfunction	4	1.5, [0.4–3.7]
	Cystitis	2	0.7, [0.1–2.6]
Reproductive tract	Endometritis/Metritis	3	1.1, [0.2, 3.2]
	Balanoposthitis	1	0.4, [0.0–2.0]
Thoracic effusion	Serohemorrhagic	45	16.6, [12.4–21.6]
	Hemothorax	13	4.8, [2.6–8.1]
	Suppurative	6	2.2, [0.8–4.8]
	Serum-fibrinous	3	1.1, [0.2, 3.2]
Abdominal effusion	Serohemorrhagic	13	4.8, [2.6–8.1]
	Hemoabdomen	4	1.5, [0.4–3.7]
	Serofibrinous	2	0.7, [0.1–2.6]
	Suppurative	1	0.4, [0.0–2.0]

**Table 6 animals-14-01852-t006:** Frequency and percentage of the most frequent histologic findings identified in the inner organs. Histologic examination was not performed on 14 animals; therefore, these data are presented considering a total of 257 animals.

Organ	Histologic Findings	n	%, [95% CI]
Lung	Lymphoplasmacytic pneumonia ± pleuritis	118	45.9, [39.7–52.2]
	Granulomatous pneumonia ± pleuritis	49	18.1, [13.7–23.2]
	Suppurative pneumonia ± pleuritis	21	8.2, [5.1–12.2]
	Neoplasia	1	0.4, [0.0–0.2]
	Pulmonary parasites	147	57.2, [50.9–63.3]
Liver	Lymphoplasmacytic (cholangio)hepatitis	60	23.3, [18.3–29.0]
	Steatosis	12	4.7, [2.4–8.0]
	Suppurative/Mixed cellular (cholangio)hepatitis ± necrosis	9	3.5, [1.6–6.5]
	Granulomatous (cholangio)hepatitis	7	2.7, [1.1–5.5]
	Necrosis/Degeneration	7	2.7, [1.1–5.5]
	Neoplasia	3	1.2, [0.2–3.4]
Intestine	Lymphoplasmacytic enteritis	42	16.3, [12.0–21.4]
	Suppurative/necrotizing enteritis	13	5.1, [2.7–8.5]
	Intestinal parasites	35	13.6, [9.7–18.4]
Spleen	Extramedullary hematopoiesis	66	25.7, [20.5–31.5]
	Reactive hyperplasia of the white pulp	39	15.2, [11.0–20.2]
	Follicular depletion/rarefaction	3	1.2, [0.2–3.4]
	Neoplasia	3	1.2, [0.2–3.4]
	Splenitis	3	1.2, [0.2–3.4]
	Necrosis	1	0.4, [0.0–0.2]
Kidneys	Lymphoplasmacytic/granulomatous/suppurative nephritis	40	15.6, [11.4–20.6]
	Degeneration/Necrosis of tubuli	4	1.6, [0.4–3.9]
	Neoplasia	2	0.8, [0.1–2.8]
Brain	Suppurative (meningo)encephalitis	12	4.7, [2.4–8.0]
	Lymphoplasmacytic (meningo)encephalitis	11	4.3, [2.2–7.5]
	Hemorrhage	7	2.7, [1.1–5.5]
	Granulomatous (meningo)encephalitis	1	0.4, [0.0–0.2]
Heart	Lymphoplasmacytic epi-/myo-/endocarditis	11	4.3, [2.2–7.5]
	Suppurative myocarditis	5	1.9, [0.6–4.5]

## Data Availability

The raw data that were analyzed in this article are available upon direct request to the authors.

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
