# Peer review of "Causes of Admission, Mortality and Pathological Findings in European Hedgehogs: Reports from Two University Centers in Italy and Switzerland"

_animals, 2024, doi:10.3390/ani14131852_

Round 1

Reviewer 1 Report

Comments and Suggestions for Authors

This is a comprehensive review covering mortality of a common European wildlife species. While the Introduction and Discussion sections are a little wordy and could be tightened up a little, most of the wordiness is geared towards presenting the importance of the study and providing context. This is nice, clinically helpful work.

Comments on the Quality of English Language

Some minor editing of English language is needed. 

Reviewer 2 Report

Comments and Suggestions for Authors

The authors prepared a manuscript entitled 'Causes of admission, mortality and pathological findings in European hedgehogs: reports from two university centres in Italy and Switzerland', addressing the important topic of the declining population of European hedgehogs. The two-centre study collected data on the causes of visits and mortality in hedgehogs in their regions, contributing to a better understanding of the health problems faced by this species.

According to the authors, the most common reason for visiting hedgehogs is damage and the most common cause of death in turn turns out to be infections.

Furthermore, the reader is provided with these data on a regional basis for two European countries. Analysing animal populations in general is a difficult task, and presenting it in this form even more so. Congratulations to the authors.

The following is my opinion and suggestions:

1. I would like to be able to compare these data with the general number of hedgehogs in that area. The reader unfortunately does not know what percentage of hedgehogs are affected by these illnesses or visits to the vet.

2. In the results section, my doubt is raised by the section with Bacteria. We do not know whether the bacteria indicated were associated with the animal's illness and the vet visit or whether they were associated with the animal's death. The same section also indicated parasites, nematodes, whose presence not in the body is not called an infection but an invasion.

Subsection "3.4.5. Detection of infectious agents" is by far the weakest point in the text. Furthermore, the section on bacterial infections is insufficiently discussed in the discussion. Please elaborate more on this topic - discuss the pathogenicity of the isolated pathogens as reported in other works. 

4. photographs need better quality presentation. Photographs should be larger with pathological lesions highlighted if we want to call this a patholomorphological lesion analysis. 

5. the presentation of histopathological findings is also unsatisfactory. Please group by type of disease entities. 

I.e. Figure 1. Neoplasia lession in lung, spleen etc. - And here present some photographs. Or vice versa, i.e: Figure 1. Lession detected in lung... - and here some photographs

Lines:

61 - wouldn't a better wording be "omnivorous species"? Instead of "generalist feeders"

173 - "(see also Statistical analysis)." what specifically are you referring to?

180 - please delete this subsection, it is not needed.

191-197 - please move to another section e.g. Statistical; as a sub-item.

208 - please put this information in the table.

210 - please insert this information in the table.

280 - viral or bacterial infections have never been a reason to visit a vet? Not every infection is fatal. 

Enterococcus - does not always lead to death - 10.2478/jvetres-2022-0020

426 - the word infection cannot refer to nematodes (invasion is appropriate). Please extract this information to another subsection. 

497 - this is a repetition from the beginning of the discussion. 

502 - please provide a citation for the second part of the sentence " In addition, considering the geographic similarities between the two countries [29] and that studies on the causes of mortality in Italian hedgehog populations are lacking, we considered this analysis fundamental, especially in the eye of possible future decline in population tion density."

Reviewer 3 Report

Comments and Suggestions for Authors

Dear Authors of the manuscript entitled “Causes of admission, mortality and pathological findings in European hedgehogs: reports of two university centers in Italy and Switzerland “, I have thoroughly read the manuscript and I found the subject interesting, scientifically correct and manuscript well written. Nevertheless I have one observation. The manuscript gives parasitic diseases as one of the major cause of death and in the first part of the text it is insisted on this fact. Later on one can read that the parasitic diseases were proven in a portion of samples, furthermore without information if this was actually the major cause of dead of infectious origin (samples were not provided for virology investigations as well). I suggest introduction infectious diseases instead of the strong statement on parasitic causes of death in European hedgehogs because it seems misleading to the reader. Is there any data on vector-borne diseases as cause of death in these animals? Since the ticks were included in another study for the analysis it can be mentioned in discussion a bit more since it seems to be important for the hedgehog population. Further, minor comments are given in the text below in points:

Line 30- I suggest clarification how are the hedgehogs found (e.g. in poor condition or in affected health status).

Line 42. Please give number how many animals were done in Italy and how many in Switzerland.

Line 43. infections diseases?

Line 62. I would avoid using … and give e.g.

Line 124 (please give abbreviation in brackets since later on La Ninna appers only)

Line 220 This sentence “The main cause of admission of hedgehogs of both countries were traumatic insults 220 (n=77, 28.4%)”, is in collision with the sentence in the abstract “primary causes of death identified were parasite-associated diseases and traumatic insults”. Furthermore, in line 256 it is later on stated “Although in both countries infectious diseases were the most frequently reported cause of mortality (n=87, 53.7% in 257 Italy; n=77, 70.6% in Switzerland), a higher percentage of animals had died due to traumatic insults in Italy (n=62, 38.3%), while only 13 animals were reported with trauma in Switzerland (11.9%)”. Please make it clear throughout the text, since infectious diseases and parasitic diseases are not the same term and it reads misleading.

Throughout the text. Please include confidence intervals

Comments on the Quality of English Language

Minor editing of English language is needed.

Round 2

Reviewer 2 Report

Comments and Suggestions for Authors

Dear authors, thank you for all your corrections.

Line 280

In Table 2 gives information on the reasons for visiting the doctor and the reasons for death. This table is in row 280. Was infection never the reason for the visit?

Best
